# COLLABORATIVE HYBRID PROPAGATOR FOR TEMPORAL MISALIGNMENT IN AUDIO-VISUAL SEGMENTATION

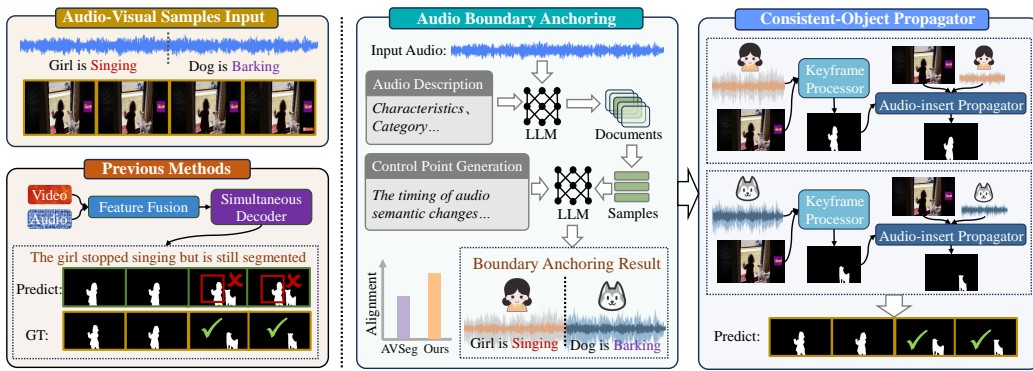

Figure 1: (a) Existing methods often show temporal misalignment between audio guidance and prediction results. (b) The Collaborative Hybrid Propagator mitigates this with two stages: Audio Boundary Anchoring and Consistent-Object Propagator.

## ABSTRACT

Audio-visual video segmentation (AVVS) aims to generate pixel-level maps of sound-producing objects that accurately align with the corresponding audio. However, existing methods often face temporal misalignment, where audio cues and segmentation results are not temporally coordinated. Audio provides two critical pieces of information: i) target object-level details and ii) the timing of when objects start and stop producing sounds. Current methods focus more on object-level information but neglect the boundaries of audio semantic changes, leading to temporal misalignment. To address this issue, we propose a **Co**llaborative Hybrid **Prop**agator Framework (Co-Prop). This framework includes two main steps: Preliminary Audio Boundary Anchoring and Frame-by-Frame Audio-Insert Propagation. To Anchor the audio boundary, we employ retrieval-assist prompts with Qwen large language models to identify control points of audio semantic changes. These control points split the audio into semantically consistent audio portions. After obtaining the control point lists, we propose the Audio Insertion Propagator to process each audio portion using a frame-by-frame audio insertion propagation and matching approach. We curated a compact dataset comprising diverse source conversion cases and devised a metric to assess alignment rates. Compared to traditional simultaneous processing methods, our approach reduces memory requirements and facilitates frame alignment. Experimental results demonstrate the effectiveness of our approach across three datasets and two backbones. Furthermore, our method can be integrated with existing AVVS approaches, offering plug-and-play functionality to enhance their performance.

# 1 INTRODUCTION

Audio-Visual Video Segmentation (AVVS) aims to generate pixel-level maps of sound-producing objects, ensuring their alignment with the corresponding audio signals. This technological advancement holds substantial promise across diverse domains, including video editing and surveillance.

Numerous studies (Li et al. 2023; Huang et al. 2023; Chen et al. 2024; Hao et al. 2023; Yang et al. 2023; Liu et al. 2024c; Rouditchenko et al. 2019; Mo & Raj 2024) have introduced innovative models and made significant contributions to the discipline. However, existing methods often suffer from **Temporal Misalignment Issue** between audio guidance and prediction outcomes. For instance, in Fig. 1 (a), a video depicts a girl and a dog. Initially, the girl sings, but later, she stops singing as the dog starts barking. Current methods frequently segment the girl's mask after she stops singing by mistake or incorrectly segment the dog's mask before it starts barking. This issue arises because existing methods do not clearly identify the time intervals when the girl sings and the dog barks.

In fact, audio as guiding information inherently comprises two crucial elements: i) object-level category information of the sound source and ii) the time points when the target object starts and stops making sound. Existing approaches tend to focus more on the category information of the sounding objects while ignoring the critical transition times. Therefore, we propose that we should first meticulously process the audio guiding information, extracting key information about the category of the sounding object and the time periods when it is making sound, and then use this guiding information to direct the video segmentation.

Thus, we introduce **Collaborative Audio Analysis and Video Segmentation**. This method begins with a detailed analysis of the audio to pinpoint key time points where the sound sources change while simultaneously obtaining object-level information. This process segments the entire audio input into multiple sub-audio segments, each maintaining consistent sound source categories over time. We then perform video segmentation on each sub-audio segment separately, thereby preemptively separating the audio of different sounding objects on the time axis to alleviate the temporal misalignment.

Additionally, to enhance the temporal alignment between the audio and prediction results, we propose **Hybrid Audio-Visual Feature Transmission**. Existing AVVS models decode all frame masks simultaneously, leading to high memory demands in long video scenarios and failing to integrate audio guiding information in a frame-aligned manner. Therefore, a hybrid audio-visual feature for frame-by-frame mask propagation is necessary.

On the whole, we propose a **Co**llaborative Hybrid **Prop**agator Framework (Co-Prop), which comprises two steps: **Preliminary Audio Boundary Anchoring** and **Frame-by-Frame Audio-Insert Propagation**. Audio Boundary Anchoring aims to meticulously extract object-level information from the audio and identify the time points where sounding objects change, referred to as control points. Specifically, we design multi-step retrieval-augmented prompts applied to the Qwen large language model. Initially, we generate descriptions and categories for the audio, then search for instances of the same category in the training set annotated with control point lists and include these as examples in the new prompt. This process yields a control point list indicating where the sounding objects change in the audio. After that, we designed the Audio-insert Propagator, which aims to perform video segmentation on the sub-audio segments. Specifically, we design a Keyframe Processor to handle keyframe predictions, fine-tuned on our curated keyframe sub-dataset. Furthermore, we design to embed audio information frame-by-frame during the propagation of keyframe masks.

Experimental results demonstrate the effectiveness of our approach across three datasets and two backbones (On M3, our 63.58% $\mathcal{M_J}$ / 73.96% $\mathcal{M_F}$ vs. AVSegFormer 58.36% $\mathcal{M_J}$ / 69.3% $\mathcal{M_F}$ with PVT-v2; On AVSS, our 39.56% $\mathcal{M_J}$ / 44.37% $\mathcal{M_F}$ vs. AVSegFormer 37.3% $\mathcal{J}$ / 42.8% $\mathcal{F}$ with PVT-v2). Our method can be integrated with existing AVVS approaches, offering plug-and-play functionality to enhance their performance. Furthermore, we curated a compact dataset comprising diverse source conversion instances and devised an assessment approach to gauge alignment efficacy. In contrast to prior methods, Co-Prop demonstrates superior alignment rates between audio and objects. Our code and benchmark will be released.

Overall, our contributions are summarized as follows:

- We observed a prevalent temporal misalignment issue between audio guidance and prediction outcomes. To address this, we propose a **Co**llaborative Hybrid **Prop**agator (Co-Prop)

framework comprising Preliminary Audio Boundary Anchoring and Frame-by-Frame Audio-Insert Propagation. Furthermore, we curated a compact dataset comprising diverse source conversion cases and devised a metric to assess alignment rates.

- We developed the Retrieval-augmented Control Points Generation Module to anchor key points during audio category transitions preemptively. Additionally, we designed the Audio-insert Propagator to embed audio frame by frame, reducing memory demands while facilitating frame-aligned integration of audio cues.

- We conduct extensive experiments on three benchmarks and achieve superior performance on all three datasets with two backbones. Furthermore, our method can be integrated with existing approaches, offering plug-and-play functionality to enhance their performance.

## 2 RELATED WORK

**Audio-Visual Video Segmentation.** The Audio-Visual Video Segmentation (AVVS) task involves using a video and its corresponding audio to generate pixel-level masks for the sounding objects. Zhou et al. introduced the AVSBench-Object (Zhou et al. 2022) and AVSBench-Semantic benchmarks (Zhou et al. 2023). Following this, several methods have made notable advancements in this field (Li et al. 2023; Huang et al. 2023; Hao et al. 2023; Chen et al. 2024; Yang et al. 2023; Liu et al. 2024b). CATR (Li et al. 2023) proposed the audio-queried transformer, which embeds audio features during the decoding stage to capture object-level information. AQFormer (Huang et al. 2023) links object queries to sounding objects and introduces the ABTI module for temporal modeling. AVSegFormer (Gao et al. 2024) employs bidirectional conditional cross-modal feature fusion to enhance audio-visual segmentation. CAVP (Chen et al. 2024) identified biases in the dataset and introduced a cost-effective strategy to address them. However, these approaches all suffer from temporal misalignment between the predictions and the audio guidance due to their failure to account for the critical transition points where the sounding objects change. To address this, we propose a two-stage solution: first, anchoring the temporal boundaries of the same target objects in the audio and then performing frame-by-frame video segmentation on the audio clips with fixed target objects.

**Video Object Segmentation**. The objective of video object segmentation (VOS) is to derive masks for target objects throughout an entire video. A prevalent approach is semi-supervised VOS, which involves segmenting a specific object with a fully annotated mask in the initial frame. Recent advancements in VOS methods have showcased innovative approaches (Lan et al. 2022; Vujasinovic et al. 2022; Yin et al. 2021; Zhu et al. 2021; Fan et al. 2021; Oh et al. 2019; Yang et al. 2021; Hao et al. 2024; Mao et al. 2023b; Liu et al. 2023a; Mao et al. 2023a). For instance, Zhu et al. 2021 proposes an approach that considers pixel-wise similarities between reference and target frames alongside the structural information of objects. Similarly, PML (Chen et al. 2018) utilizes the nearest neighbor classifier to learn pixel-wise embedding, OGS (Fan et al. 2021) introduces an architecture based on object-aware global-local correspondence. STM (Oh et al. 2019) employs a memory bank constructed from past frames, while AOT (Yang et al. 2021) introduces an identification mechanism to process multiple objects within a frame. However, these methods only consider video features during propagation and do not incorporate audio guidance. To address this, we designed the Audio-insert Propagator, trained on audio-visual datasets, to integrate audio guidance information during frame-by-frame propagation. This enhancement allows the prediction process to consider audio cues.

## 3 METHOD

### 3.1 OVERVIEW

Our pipeline comprises two components: the Retrieval-augmented Control Points Generation Module (RCPG) and the Audio-insert Propagation Module (AIP). The RCPG module anchors key points of audio object category transitions, segmenting the audio into sub-clips with consistent target objects. The AIP module performs video segmentation for each sub-audio clip. Compared to the existing methods that process audio and all video frames simultaneously, our approach reduces memory requirements and enhances temporal alignment between audio and prediction results.

**Retrieval-Augmented Control Points Generation Module.** The RCPG module detects key points where the sounding objects change in the audio, referred to as control points. Based on these control

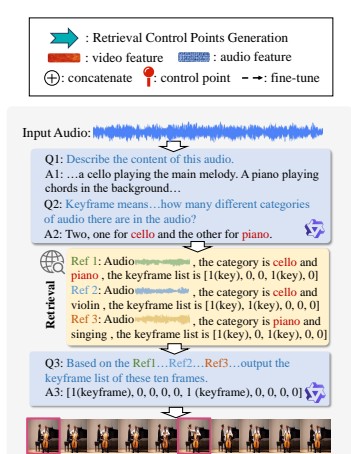 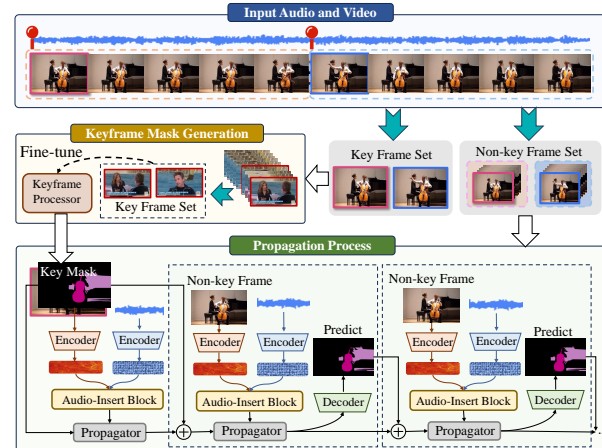

(a) Retrieval-Augmented Control Points Generation

(b) Audio-Insert Propagator Module

Figure 2: **Overview**. Our Collaborative Hybrid Propagator Framework (Co-Prop) comprises Retrieval-Augmented Control Points Generation and Audio Insertion Propagation. Retrieval-augmented Control Points Generation Module aims to anchor key points during audio category transitions preemptively. Additionally, the Audio-insert Propagator aims to embed audio frame by frame, reducing memory demands while facilitating frame-aligned integration of audio cues.

points, the audio is divided into sub-audio segments. In each segment, the first frame is designated as the keyframe, while the remaining frames are considered normal frames,

$$[c_i]_{i=0}^T = RCPG(A, x), c_i \in \{0, 1\}, \tag{1}$$

where we denote the Retrieval-augmented Control Points Generation Module as $RCPG(\cdot)$, with $A$ representing the audio, $x$ representing the prompt, and $T$ denoting the number of frames. The $c_i$ serves as the flag for control points. A value of 1 for $c_i$ designates the associated video frame as a key video frame $V^{key}$ and the audio frame $A^{key}$ as a key audio frame. Conversely, a value of 0 for $c_i$ indicates that the corresponding video frames $V^{normal}$ and audio frames $A^{normal}$ are normal frames.

**Audio-Insert Propagation.** The goal of Audio-Insert Propagation is to process the sub-audio segments. Thus, we designed the Keyframe Processor module to handle the keyframes and the Audio-insert Propagator to handle the normal frames.

To carefully leverage audio guidance information to identify the target objects accurately, we have developed a dedicated single-frame image segmentation model tailored for keyframes,

$$M^{key} = KeyPro(V^{key}, A^{key}), \tag{2}$$

where $KeyPro(\cdot)$ denotes the Keyframe Processor, handles video and audio keyframes, represented as $V^{key}$ and $A^{key}$ respectively. Subsequently, we derive the keyframe mask $M^{key}$.

After acquiring masks for keyframes from the Keyframe Processor, we employ a mask propagation technique to derive the masks for normal frames $M^{normal}$. Unlike the existing methods (Oh et al. 2019; Yang et al. 2021) that solely considered video features, our Audio-Insert Propagator is designed to embed audio information frame by frame into the mask propagation based on $M^{key}$,

$$M^{normal} = AudioProp(M^{key}, V^{normal}, A^{normal}), \tag{3}$$

where $AudioProp(\cdot)$ denotes the Audio-Insert Propagator, $V^{normal}$ and $A^{normal}$ denote the video normal frames and audio normal frames. Additionally, $M^{key}$ signifies the mask of keyframes. Finally, we derive masks for all normal frames, obtaining the complete video masks.

### 3.2 RETRIEVAL-AUGMENTED CONTROL POINTS GENERATION

Existing AVVS methods process audio and video features together through feature fusion, which hinders the effective extraction of control points where target objects change, leading to temporal

misalignment between the audio and prediction results. To mitigate this issue, we propose preemptively anchoring the boundary of target object transitions. Leveraging the Qwen large language model (LLM), which excels in audio processing, we designed multi-step retrieval prompts to generate control points for the corresponding audio. This approach identifies the boundary of sound-producing object transitions, dividing the audio into sub-audio clips with consistent target objects.

**LLMs-Based Audio Description.** Qwen LLM excels in audio processing, making it suitable for handling our input audio, thus the RCPG module operates without requiring additional training. Initially, we input the audio and a prompt into the LLM to generate the audio description. As shown in Fig. 2 (a), we used the simple prompt "Please briefly describe the content of this audio." We then received the corresponding audio description: "This is a live performance of a classical music piece. A cello plays the main melody and a piano plays chords in the background. The atmosphere is sentimental." This demonstrates the keen perception of LLM and shows that a descriptive prompt helps the model initially understand the audio.

Our next goal was to obtain the category information of the sounding objects. To achieve this, we designed a second prompt as shown in Fig. 2 (a). The response was: "There are two different types of audio in this segment, one for cello and the other for piano." Consequently, we identified two target objects for segmentation in this audio: the cello and the piano.

We manually annotated the audio categories and control points in the training set for audio-video segmentation. After identifying the audio category to be predicted, we search the training set for samples of the same category. Since the sounding objects of the same category exhibit certain temporal similarities, we utilize the control points from these existing samples as additional knowledge to aid the LLM in learning and making accurate judgments.

$$D_q = R(q, D), q = Qwen(\bar{x}, A), \tag{4}$$

where $Qwen(\cdot)$ denotes the Qwen LLM, $\bar{x}$ represents the audio description category prompt, $A$ denotes the audio to be predicted, from which we derive the audio category $q$. $R(\cdot)$ also signifies the retrieval function, and $D$ represents the training set document with annotated control points. Consequently, we obtain samples of the same category $D_q$.

**Example-Based Retrieval.** To anchor the pivotal time points of target-object transitions in the audio, we designed prompt $x$ with the annotated samples. Firstly, we defined control points as follows: "When the category, timbre, and quantity of the current frame audio differ from the previous frame, we call the current frame a keyframe." Additionally, we provided the number of video frames and requirements for the generated results: "The audio frames are evenly divided into ten frames, keyframes are marked as 1, and non-key frames are marked as 0. Please output the categories of these ten frames in order from the first frame to the tenth frame in the format of a list." Finally, we obtained the control points list corresponding to the audio.

$$[c_i]_{i=0}^T = Qwen([D_q, x], A), \tag{5}$$

where $Qwen(\cdot)$ denotes the Qwen LLM, $D_q$ represents the samples of the same category, $x$ signifies the designed prompt, $A$ refers to the input audio, and $[c_i]_{i=0}^T$ represents the control points list corresponding to the audio. We then divide the audio into several sub-audio segments based on the control points list and employ the Keyframe Processor and Audio-Insert Propagator to perform video segmentation on these sub-audio segments.

## 3.3 KEYFRAME PROCESSOR

We divided the audio into several sub-audio segments based on the control points list obtained from the Retrieval-Augmented Control Points Generation module. We designated the first frame of each sub-audio segment as the keyframe and designed a Keyframe Processor (KPF) to obtain masks for these keyframes, laying the foundation for the subsequent propagation of non-key frames. Furthermore, we fine-tuned the Keyframe Processor on a restructured keyframe training dataset.

**Keyframe Mask Generation.** Keyframe processor is an audio-guided image segmentation model designed to generate masks for keyframes from their corresponding audio and images. It operates by first extracting image features and audio features from the key frames. These features are then integrated through cross-attention mechanisms at each layer. Then we apply the audio-queried decoding (Li et al. 2023) to process the integrated features and produce the keyframe masks.

Concretely, video frames and audio frames are extracted at predefined control points first. Then these frames are encoded to derive the video features $F_v^{key} \in \mathbb{R}^{T^k \times H \times W \times C}$ and audio features $F_a^{key} \in \mathbb{R}^{T^k \times C}$, $T^k$ denotes the number of keyframes. The video features are then flattened into dimensions represented by $H \times W$ and combined with the audio features through concatenation. The resulting concatenated features are fed into a transformer encoder, enabling the integration of audio features with the video features specific to keyframes. Finally, the fused video feature set undergoes decoding in a dedicated decoder module to generate the masks corresponding to the keyframes.

**Keyframes Dataset Collection and Fine-tuning.** The Keyframe Processor is tailored for the keyframe image analysis. Initially, all data from the training set was leveraged during the initial training phase. To optimize the Processor's performance on keyframes, we adopted the retrieval-augmented control points generation method (Sec 3.2) to annotate keyframes within the training set, thereby generating a specialized subset dedicated to keyframes. Following this, the Keyframe Processor was fine-tuned exclusively on this subset.

## 3.4 AUDIO-INSERTED PROPAGATOR

Existing propagation methods (Oh et al. 2019; Yang et al. 2021; Heo et al. 2020; 2021; Li et al. 2024; Rajič et al. 2023) rely solely on video features to propagate masks without considering guidance from audio. To better incorporate audio information, we designed the Audio-Insert Propagator to embed audio frame-by-frame during the propagation of keyframe masks.

**Audio Insertion.** We integrate audio features with the image features of the current frame before propagation. The image features are processed through four layers, with the fourth layer used to fuse with the audio features. Unlike existing propagation method (Yang et al. 2021), which relies solely on mask-generated identities for object markers during propagation and matching, our method combines audio features and masks to generate these identities.

During the propagation process, we input the current video frame into an encoder and obtain the four layers of video frame features denoted as $F_v^l \in \mathbb{R}^{T^k \times C \times H \times W}$. Simultaneously, we process the audio features of the current frame through an audio encoder, obtaining feature $F_a \in \mathbb{R}^{T^k \times D}$. Subsequently, we employ cross-attention (Chen et al. 2021) to embed the audio feature,

$$\tilde{F}_v^l = AudioInsert(F_v^l, F_a) = \text{Softmax}\left(\frac{F_v^l W^Q \cdot \left(F_a W^K\right)^{\text{T}}}{\sqrt{d_{\text{head}}}}\right) F_a W^V, \tag{6}$$

where the $AudioInsert(\cdot)$ denotes the Audio-Insert Block. In $AudioInsert(\cdot)$, the query is the video feature $F_v^l$, and key is the audio feature $F_a$. Moreover, $W^Q, W^K, W^V \in \mathbb{R}^{C \times d_{\text{head}}}$ are learnable parameters. Consequently, we obtain the current video frame features $\tilde{F}_v^l$ embedded with the guidance information from the current audio feature.

**Propagation Process.** The Propagator Block (Yang et al. 2021) begins with a self-attention layer to learn associations among targets within the current frame. It then incorporates long-term attention to aggregate information from memory frames and short-term attention to capture temporal smoothness from adjacent frames. The final component consists of a 2-layer feed-forward MLP with GELU non-linearity. All attention modules are implemented using multi-head attention, which involves multiple attention mechanisms followed by concatenation and a linear projection.

Specifically, we input the video features, enriched with audio features of the current frame, into the Propagator Block. Additionally, memory information from preceding video frames and corresponding masks predicted from earlier frames are incorporated, facilitating the derivation of predictive insights for the current frame,

$$E^t = Propagator(\tilde{F}_v^{l,t}, E^{t-1}, M^{t-1}), \tag{7}$$

where $Propagator(\cdot)$ represents the Propagator Block, while $\tilde{F}_v^{l,t}$ signifies the $l$-th layer video feature of the current frame at time $t$. $E^{t-1}$ denotes information from preceding frames, and $M^{t-1}$ indicates the predicted mask from the previous frame. Following the acquisition of the current frame's embedding $E^t$, it is fed into the decoder for the prediction of the frame's mask $M^t$.

Table 1: Quantitative comparisons on AVSBench-object datasets (single-source, S4; multi-source, M3) and AVSBench-semantic dataset (AVSS).

| Method | | | Backbone | S4 | | M3 | | AVSS | |
|---|---|---|---|---|---|---|---|---|---|
| | | | | $\mathcal{M}_{\mathcal{J}}$ | $\mathcal{M}_{\mathcal{F}}$ | $\mathcal{M}_{\mathcal{J}}$ | $\mathcal{M}_{\mathcal{F}}$ | $\mathcal{M}_{\mathcal{J}}$ | $\mathcal{M}_{\mathcal{F}}$ |
| TPAVI | Zhou et al. 2022 | *ECCV'2022* | *ResNet* | 72.8 | 84.8 | 47.9 | 57.8 | 20.2 | 25.2 |
| | | | *PVT-v2* | 78.7 | 87.9 | 54.0 | 64.5 | 29.8 | 35.2 |
| CATR | Li et al. 2023 | *ACM MM'2023* | *ResNet* | 74.8 | 86.6 | 52.8 | 65.3 | 23.4 | 28.6 |
| | | | *PVT-v2* | 81.4 | 89.6 | 59.0 | 70.0 | 32.8 | 38.5 |
| AuTR | Liu et al. 2023b | *arXiv'2023* | *ResNet* | 75.0 | 85.2 | 49.4 | 61.2 | - | - |
| | | | *PVT-v2* | 80.4 | 89.1 | 56.2 | 67.2 | - | - |
| AQFormer | Huang et al. 2023 | *IJCAI'2023* | *ResNet* | 77.0 | 86.4 | 55.7 | 66.9 | - | - |
| | | | *PVT-v2* | 81.6 | 89.4 | 61.1 | 72.1 | - | - |
| BAVS | Liu et al. 2024a | *TMM'2024* | *ResNet* | 78.0 | 85.3 | 50.2 | 62.4 | 24.7 | 29.6 |
| | | | *PVT-v2* | 82.0 | 88.6 | 58.6 | 65.5 | 32.6 | 36.4 |
| AVS-BiGen | Hao et al. 2023 | *AAAI'2024* | *ResNet* | 74.1 | 85.4 | 45.0 | 56.8 | - | - |
| | | | *PVT-v2* | 81.7 | 90.4 | 55.1 | 66.8 | - | - |
| AVSegFormer | Gao et al. 2024 | *AAAI'2024* | *ResNet* | 76.4 | 86.7 | 53.8 | 65.6 | 26.6 | 31.5 |
| | | | *PVT-v2* | 82.1 | 89.9 | 58.36 | 69.3 | 37.3 | 42.8 |
| Ours | | | *ResNet* | **78.5** | **87.2** | **57.2** | **68.4** | **30.2** | **35.8** |
| | | | *PVT-v2* | **83.7** | **90.9** | **63.6** | **74.0** | **39.6** | **44.4** |

# 4 EXPERIMENT

## 4.1 COMPARISON

We evaluated the performance of the proposed framework on three datasets using two backbones. Overall, our method achieved significant improvement, particularly on the M3 and AVSS datasets.

From the results in Table 1, we have the following observations:

1) **Mitigating the temporal misalignment issue**. Our study demonstrates that our model exhibits more substantial performance enhancements on the M3 and AVSS datasets than the S4 dataset. This discrepancy arises from the multi-source audio nature of the M3 and AVSS datasets, which encompass diverse sound sources, exacerbating temporal misalignment between audio and predictions. We conducted a comparative analysis of alignment rates on the MOC dataset. We introduce the Alignment Rate metric, representing the proportion of predicted video frames where the identified object aligns with the ground truth, to the total number of frames assessed. We compared Alignment Rate results using the MOC dataset (see Fig. 3). Our model demonstrates more pronounced perfor-

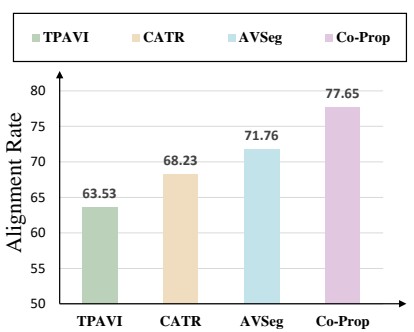

Figure 3: Comparison Alignment Rate on MOC Test Dataset.

mance enhancements on datasets with multi-source audio, which involves segmenting audio intervals based on transitions in audio-producing objects, followed by audio-visual segmentation within each interval, thereby effectively mitigating this challenge.

2) **Mitigating the pixel-level contour issue.** Fig. 5 shows that previous methods often produce inaccurate edge contours in pixel-level segmentation predictions. In contrast, our method significantly improves the delineation of target object contours. This improvement is attributed to our use of a frame-by-frame propagation method within segmented sub-audio clips where the target object remains unchanged and the AOT-Large pre-trained model, trained on a large-scale dataset. Consequently, our model excels in detecting and tracking the position and contours of target objects.

Table 2: Ablation Study on M3 and S4 datasets with Pvt-v2 backbone.

(a) Ablation Study of Main Modules

| KPF | RCPG | AIP | M3 | | S4 | |
|---|---|---|---|---|---|---|
| | | | $\mathcal{M}_{\mathcal{J}}$ | $\mathcal{M}_{\mathcal{F}}$ | $\mathcal{M}_{\mathcal{J}}$ | $\mathcal{M}_{\mathcal{F}}$ |
| ✘ | ✘ | ✘ | 58.63 | 69.71 | 79.51 | 88.24 |
| ✔ | ✘ | ✘ | 58.82 | 70.05 | 80.89 | 89.03 |
| ✔ | ✔ | ✘ | 61.97 | 72.13 | 82.21 | 89.94 |
| ✔ | ✔ | ✔ | 63.58 | 73.96 | 83.71 | 90.86 |

(b) Ablation Study of on RCPG Sub-Modules

| Method | M3 | | S4 | |
|---|---|---|---|---|
| | $\mathcal{M}_{\mathcal{J}}$ | $\mathcal{M}_{\mathcal{F}}$ | $\mathcal{M}_{\mathcal{J}}$ | $\mathcal{M}_{\mathcal{F}}$ |
| Cosine | 59.21 | 70.45 | 81.22 | 89.13 |
| 1-step | 61.05 | 70.83 | 81.45 | 89.22 |
| 3-step | 61.85 | 71.22 | 81.72 | 89.31 |
| RCPG | 63.58 | 73.96 | 83.71 | 90.86 |

## 4.2 EXPERIMENT SETUP

**Datasets.** We evaluated our method on three benchmarks: 1) **M3** (Zhou et al. 2022) (Fully-supervised Multiple-sound Source Segmentation). M3 datasets provide binary segmentation maps identifying the pixels of sounding objects, and each example of M3 contains multiple sources of audio. 2) **S4** (Zhou et al. 2022) (Semi-supervised Single-sound Source Segmentation). S4 datasets also provide binary segmentation maps identifying the pixels of sounding objects, and each example of S4 contains single sources, supplying ground truth solely for the initial frame during training. 3) **AVSS** (Zhou et al. 2023) (Fully-supervised Audio-Visual Semantic Segmentation). The AVSS dataset offers semantic segmentation maps as labels. 4) **MOC** (Multiple-sound Source Conversion). From the original M3 data test set of 64 examples, we selected 17 instances featuring multiple target objects that change over time to create the MOC test set. Consequently, the MOC test set provides a more rigorous evaluation of the model's ability to synchronize audio inputs with corresponding predictions over time as predicting dynamic, multi-object scenarios is inherently complex.

**Training Details.** Our system is structured in a two-stage training process. In the first stage, we train the Keyframe Processor Network and fine-tune it on a keyframe dataset collected. The image feature extraction backbones are ResNet-50 (He et al. 2016) and Pyramid Vision Transformer (PVT-v2) (Wang et al. 2021), while the VGGish model (Hershey et al. 2017) is employed for audio feature extraction. We use the Adam optimizer with a learning rate of 1e-4 and trained for 100 epochs with a batch size of 4. The model was trained on a 40G A100. Notably, the Keyframe Processor is interchangeable with other audio-visual segmentation models, whose predictions can be input into our Audio-insert Propagator for enhanced performance.

Table 3: The MOC test dataset we proposed is essential for advancing the evaluation of misalignment issues in audio-visual synchronization. Avg.Cate denotes the average number of categories per video, and Category Changes denotes the proportion of videos with audio category changes.

| Dataset | Videos | Avg.Cate | Category Changes |
|---|---|---|---|
| S4 | 740 | 1 | 0% |
| M3 | 64 | 1.375 | 26.56% |
| MOC | 17 | 2.176 (↑ 0.801) | 100% (↑ 73.44%) |

In the second stage, we train the Audio-insert Propagator on the S4, M3, and AVSS datasets using the pre-trained AOT-Large model (Yang et al. 2021) with a ResNet-50 backbone. This stage integrates four layers of video features into the Audio Embedder, with channels as 256.

**Evaluation Metrics.** We employed the standard evaluation metrics Jaccard index ($\mathcal{J}$) (Everingham et al. 2010) and F-score ($\mathcal{F}$) in our experiments. The mean values over the entire dataset are $\mathcal{M}_{\mathcal{J}}$ and $\mathcal{M}_{\mathcal{F}}$. $\mathcal{M}_{\mathcal{J}}$ quantifies the intersection-over-union between the predicted segmentation mask and the ground-truth mask, while $\mathcal{M}_{\mathcal{F}}$ assesses the balance between precision and recall. Moreover, we introduce the Alignment Rate metric, representing the proportion of predicted video frames where the identified object aligns with the ground truth, to the total number of frames assessed.

## 4.3 ABLATION STUDY

Table 2 (a) presents the results of ablation experiment on the three main modules. The baseline follows our proposed two-stage processing approach: audio boundary anchoring and video segmentation corresponding to sub-audio clips. In the baseline model, we use cosine similarity to obtain the control

Table 4: Comparative experiments to evaluate the effectiveness of replacing the video feature with prediction text in propagation.

| Method | M3 | | S4 | |
|---|---|---|---|---|
| | $\mathcal{M}_{\mathcal{J}}$ | $\mathcal{M}_{\mathcal{F}}$ | $\mathcal{M}_{\mathcal{J}}$ | $\mathcal{M}_{\mathcal{F}}$ |
| Co-Prop (Direct-guided) | 63.6 | 74 | 83.7 | 90.9 |
| Text-guided | 52.8($\downarrow$10.8) | 63.7($\downarrow$10.3) | 78.5($\downarrow$5.2) | 87.1($\downarrow$3.8) |

point list and employ the original AOT (Yang et al. 2021) within the sub-audio clips. From the results in Table 2 (a), we have the following observations:

1) The model's performance relies more on the Keyframe Processor when the performance of audio boundary anchoring is not good enough. Table 2 (a) shows that fine-tuning the Keyframe Processor on our curated keyframe dataset improves its performance in audio-visual segmentation, enabling more accurate prediction of keyframe masks and thereby enhancing overall model performance.

2) Improving the accuracy of audio boundary anchoring can boost overall model performance. We used the Qwen LLM, and we designed novel multi-step retrieval prompts. Compared to simply using cosine similarity, the RCPG module has better ability to anchor the boundaries of audio transitions, thereby improving model performance.

3) Introducing audio guidance information is essential when performing video segmentation on sub-audio clips. The original AOT method cannot embed audio guidance information. Our designed Audio-insert Propagator embeds audio guidance information frame by frame and trains it on corresponding audio-visual segmentation datasets, thus enhancing performance during the audio-visual segmentation propagation phase.

**Ablation Study of Main Modules.** Table 2 (b) presents the ablation study on RCPG Sub-Modules, which investigates various model variants for the RCPG module. We designed the following variants: 1) 1-Step: This variant utilizes a single prompt to generate a list of control points. The prompt instructs: "Divide the audio into five frames. Assume the audio category of the first frame is 1. If the category of the current frame matches the previous frame, output 0; otherwise, output 1. Provide the categories of frames one through five in sequence, formatted

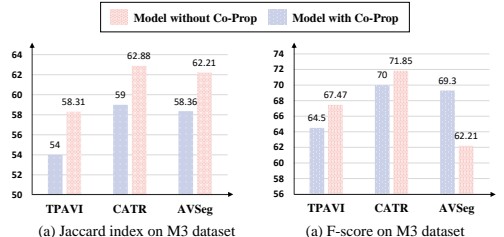

Figure 4: Comparison w/o Co-Prop. Pink denotes the model with Co-Prop as Keyframe Processor.

as a list." 2) 3-Step: This method employs three sequential prompts without supplementary reference information. First, we prompt Qwen to describe the audio features with: "Please describe the input audio." Next, we inquire: "How many different sound categories are present in the audio?" Finally, we request Qwen to generate the control points list based on audio changes, using the prompt from the 1-Step approach. 3) RCPG Module: This variant builds on the 3-Step approach by integrating additional reference information from the training set, specifically preprocessing the ground truth mask into corresponding control points lists, which are included as reference examples in the prompt.

The results presented in Table 2 (b) yield several key observations: 1) Multi-step prompts facilitate progressive thinking in the model, enhancing overall reasoning performance compared to single-step prompts. On M3, "3-step" 61.85% $\mathcal{M}_{\mathcal{J}}$ / 71.22% $\mathcal{M}_{\mathcal{F}}$ vs. "1-step" 61.05% $\mathcal{M}_{\mathcal{J}}$ / 70.83% $\mathcal{M}_{\mathcal{F}}$; On S4, "3-step" 81.72% $\mathcal{M}_{\mathcal{J}}$ / 89.31% $\mathcal{M}_{\mathcal{F}}$ vs. "1-step" 81.45% $\mathcal{M}_{\mathcal{J}}$ / 89.22% $\mathcal{M}_{\mathcal{F}}$. 2) The inclusion of relevant training set samples as reference content for prompts clarifies the guiding information, providing the model with more reliable reference samples for reasoning, thereby improving overall performance. On M3, "RCPG" 63.58% $\mathcal{M}_{\mathcal{J}}$ / 73.96% $\mathcal{M}_{\mathcal{F}}$ vs. "3-step" 61.85% $\mathcal{M}_{\mathcal{J}}$ / 71.22% $\mathcal{M}_{\mathcal{F}}$; On S4, "RCPG" 83.71% $\mathcal{M}_{\mathcal{J}}$ / 90.86% $\mathcal{M}_{\mathcal{F}}$ vs. "3-step" 81.72% $\mathcal{M}_{\mathcal{J}}$ / 89.31% $\mathcal{M}_{\mathcal{F}}$.

**Ablation Study of on RCPG Sub-Modules.** *1) Settings.* Table 2 (b) presents various design schemes for the Retrieval-Augmented Control Points Generation (RCPG) sub-module. In this experiment, the fine-tuned Keyframe Processor was used to manage keyframes at the control points, and the Audio-insert Propagator was employed to propagate sub-audio clips. We compared three prompt design methods, specifically investigating the effects of prompt step sizes and the impact of retrieval assist. In prompt design, using a 3-step dialogue yields better performance than using a step size of

Figure 5: Comparative analysis of the AVSeg method and our proposed model. We present three qualitative examples from the M3 datasets. The samples illustrates the effective performance of Co-Prop in addressing temporal misalignment and pixel-level contour issues.

one when the content is identical. Additionally, we enhance the prompt by incorporating samples with the control point list from annotated instances of the same audio category identified by LLM in the training set. This Retrieval-assist method significantly improves the feedback quality of the LLM.

**Can Text Labels Replace Audio for Guiding Video Segmentation?** We explored the possibility of converting audio directly into text labels to guide image segmentation. However, this approach tends to accumulate significant errors. We conducted comparative experiments to evaluate the effectiveness of using text labels derived from audio categories identified by Qwen for guiding video segmentation, see Table 4. The results indicate that using text labels for guidance degrades the model's performance.

The experiments reveal that the performance drop is due to the amplification of segmentation errors caused by incorrect labels. Given the semantic ambiguity of audio, many objects produce similar sounds. For example, if a cat's meow is misclassified as "a child crying," and this label is used for segmentation, the model may produce empty predictions if no children are present in the video, significantly exacerbating the error. In contrast, our designed Keyframe Processor effectively mitigates the issue of error amplification. Compared to using text labels directly, when object sounds are very similar, the Keyframe Processor can consider both image and audio information to correct for target objects, thereby avoiding the issue of arbitrarily predicting empty masks.

## 5 CONCLUSION

We introduce a novel Collaborative Hybrid Propgator that can be integrated with existing AVVS approaches, offering plug-and-play functionality to enhance their performance. To mitigate the temporal misalignment issue that commonly exist in previous methods, we propose preliminary audio boundary anchoring. Concretely, we designed a retrieval-augmented control points generation module, applying retrieval prompts to an LLM to preemptively anchor the time points of sounding object changes, thereby alleviating the temporal misalignment issue. We designed a Keyframe Processor to obtain masks for these keyframes, laying the foundation for the subsequent propagation of non-key frames. Furthermore, we developed an audio insertion propagation module that embeds audio information frame by frame during mask propagation, which not only reduces memory requirements but also allows for frame-aligned consideration of audio guidance.

**Limitations:** Our framework remains reliant on the performance of the Keyframe Processor. If the Keyframe Processor yields poor results, the final prediction will be compromised.

**Broader Impact:** We address the core challenge of audio-video alignment in audio-guided video segmentation by proposing a novel two-stage approach. The innovative framework of Co-Prop allows for modularization and performance enhancement. Its superior performance makes Co-Prop valuable for highlighting objects in augmented and virtual reality environments, as well as for generating pixel-level object maps for surveillance inspection. We anticipate our research will contribute to advancing the practical applications of audio-guided video segmentation.

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

# A  APPENDIX

## A.1  DETAILS OF THE RCPG MODULE INFERENCE

**The Annotation of the Control Points List** $D$**.** The ground truth (GT) masks in the training set provide information on changes in sound-emitting objects. We process these GT masks to generate control points annotations. Specifically, we assess changes in the target objects by examining the consistency of semantic information across consecutive frame masks. A frame is deemed a key frame if the semantic information of the objects in the GT masks of consecutive frames varies.

---

**Algorithm 1** RCPG Inference Process

---

**Input:** Audio $A$, Prompt $x$
**Output:** $[c_i]_{i=0}^T$, control points list corresponding to the audio $A$
  1: Convert the GT masks of the training set into control points lists $D$;        ▷ *Preprocessing*
  2: Use the prompt $Q1$ to describe the content of the audio.;       ▷ *Step-by-step inference*
  3: Use prompt $Q2$ to identify the category $q$ of the audio;
  4: Retrieve the samples and control points lists $D_q$ from the training set $D$ based on the category $q$;
  5: Take the samples $D_q$ as a prompt and input it along with the audio $A$ into Qwen through Eq.(5)
  6: return the control points list $[c_i]_{i=0}^T$                  ▷ *Final prediction*

---

## A.2  ABLATION ANALYSIS ON AUDIO-INSERT PROPAGATOR SUB-MODULES

*1) Settings.* Table 5 illustrates the impact of varying the number of video feature layers in the Audio-insert Propagator module on model performance. The designed Audio Embedder module comprises four layers of video features. We explored two audio embedding methods: the first involves interacting audio features with video features with $C = 256$, while the second method engages audio features with four layers of video features, with $C = [24, 32, 96, 256]$.
*2) Results.* Compared to AOT, the Audio-insert Prop-

Table 5: Ablation on Audio-insert Propagator.

| Method | M3 | | S4 | |
|---|---|---|---|---|
| | $\mathcal{M}_{\mathcal{J}}$ | $\mathcal{M}_{\mathcal{F}}$ | $\mathcal{M}_{\mathcal{J}}$ | $\mathcal{M}_{\mathcal{F}}$ |
| AOT | 62.97 | 72.13 | 82.21 | 89.94 |
| 1-layer | 63.25 | 73.31 | 82.97 | 90.47 |
| 4-layer | 63.58 | 73.96 | 83.71 | 90.86 |

agator can embed audio guidance information frame by frame and has been trained on audio-visual datasets. Consequently, even single-layer audio-visual feature interaction enhances model performance. Furthermore, experiments demonstrate that four-layer audio-visual feature interaction is more comprehensive than single-layer interaction, leading to a significant performance boost.

## A.3  THE RESPECTIVE RESULTS FOR KEYFRAMES AND NORMAL FRAMES.

We tested the performance of Co-Prop on key frames and normal frames separately. The experimental results indicate that Co-Prop performs better on normal frames than on key frames. This improved performance can be attributed to two primary factors: (1) The Audio-Insert Propagator Module is built on AOT that demonstrates enhanced capability in boundary detection and object completion during video segmentation. (2) Our audio-insert method integrates audio

Table 6: Comparative experiments to evaluate the effectiveness of using text-guided and audio-guided.

| Data | M3 | | S4 | |
|---|---|---|---|---|
| | $\mathcal{M}_{\mathcal{J}}$ | $\mathcal{M}_{\mathcal{F}}$ | $\mathcal{M}_{\mathcal{J}}$ | $\mathcal{M}_{\mathcal{F}}$ |
| All Frames | 63.58 | 73.96 | 83.71 | 90.86 |
| Key Frames | 59.82 | 70.59 | 79.25 | 86.55 |
| Normal Frames | 65.19 | 75.39 | 85.03 | 92.37 |

features frame-by-frame with image features, facilitating more accurate audio-guided instruction.

Furthermore, there is considerable potential for enhancing the keyframe processor's performance. Employing a more advanced model for the keyframe processor could boost overall performance.