# OpenReview forum: "Collaborative Hybrid Propagator for Temporal Misalignment in Audio-Visual Segmentation"
_ICLR.cc/2025/Conference — ICLR 2025 Conference Withdrawn Submission_

### Official Review · Reviewer_G4kD · 2024-10-28

**Soundness:** 3
**Presentation:** 3
**Contribution:** 3
**Rating:** 8
**Confidence:** 5

**Summary:**

To address the temporal misalignment issue in the audio-visual video segmentation task, this paper proposes a novel two-stage method. In the first stage, the authors design modules for preliminary audio boundary anchoring. Key audio frames (control points) and corresponding visual frames reflecting the audio transition are obtained. Subsequently, in the second stage, the authors propose an audio-insert propagator module to generate pixel-level segmentation maps for normal frames by propagating masks from key frames. Experiments on three sub-benchmarks demonstrate the effectiveness and superiority of the proposed method.

**Strengths:**

1. Existing methods for audio-visual video segmentation tend to design increasingly complex architectures to enhance audio-visual interactions or pixel-level segmentation results. Unlike them, this paper highlights the temporal misalignment issue between audio and visual signals, which is a valuable research direction for the AVVS problem.
2. The proposed method, especially the preliminary audio boundary anchoring module in the first step, seems to be interesting and well-motivated.
3. The experimental results are extensive, and the proposed method achieves significant performance improvement.

**Weaknesses:**

1. Given the aforementioned strengths, I am basically satisfied with this work. A potential weakness could be that the ideas of key frame anchoring (first stage) and mask propagation (second stage) likely originate from existing methods in video object segmentation, despite these contributions being seamlessly integrated into the audio-visual video segmentation task.
2. Although the proposed method achieves satisfactory performance on three datasets, the authors only compared several baselines in Table 1. A more comprehensive review of recently published works for AVVS is required. Moreover, it would be better to discuss with some typical methods from other audio-visual learning tasks, such as audio-visual event localization and video parsing, in the related work section.

**Questions:**

1. In Figure 2(a), only the audio signal is used for control point generation. Is it possible to also introduce the visual frames in this process? (In my opinion, this would be more beneficial for the LLM to better classify the correct sounding objects). Are there suitable metrics or potential measures to evaluate/guarantee the correctness of the generated object categories, as many objects have similar sounds?
2. The authors rely on the Qwen LLM to generate control points. Have the authors tried other superior LLM models?
3. The proposed method consists of two stages. Would it be possible to consider an end-to-end strategy to address the temporal misalignment issue?

---

### Official Review · Reviewer_VUir · 2024-11-01

**Soundness:** 3
**Presentation:** 3
**Contribution:** 2
**Rating:** 5
**Confidence:** 3

**Summary:**

This paper addresses temporal misalignment issues in audio-visual video segmentation (AVVS), where current methods fail to synchronize audio cues and segmentation outputs. The proposed solution, the Collaborative Hybrid Propagator Framework (Co-Prop), includes two main components: Retrieval-Augmented Control Points Generation (RCPG) and the Audio-Insert Propagator. The RCPG module anchors the audio's temporal boundaries by leveraging a large language model (Qwen) to generate control points, splitting the audio into semantically consistent segments. The Audio-Insert Propagator performs frame-by-frame video segmentation, embedding audio guidance information to align audio cues with video frames effectively.

**Strengths:**

1. Novelty in Addressing Temporal Misalignment: The two-stage Co-Prop framework innovatively tackles the temporal misalignment issue. By anchoring the temporal boundaries and inserting audio cues frame-by-frame, the model effectively improves synchronization between audio and visual data.
2. The paper provides extensive experimental evidence demonstrating the framework's effectiveness on multiple datasets (S4, M3, and AVSS) and backbones (ResNet and PVT-v2).
3. Introducing the MOC test set and a new alignment rate metric to measure synchronization accuracy.

**Weaknesses:**

1. Reliance on Qwen LLM: With many multimodal models (MLMs) now capable of video understanding(Gemini, Video-llama, ...) and even reasoning segmentation(LISA), what specific advantages does the proposed approach offer over directly using these large models?
2. Accuracy Discrepancy for AVSegFormer: The reported accuracy for AVSegFormer in this paper doesn’t align with that in the cited references.

**Questions:**

See the weakness part.

---

### Official Review · Reviewer_jhjq · 2024-11-02

**Soundness:** 3
**Presentation:** 3
**Contribution:** 3
**Rating:** 5
**Confidence:** 4

**Summary:**

The paper presents the Co-Prop, a new framework that addresses temporal misalignment in audio-visual video segmentation by enhancing alignment between audio cues and visual segmentation outputs. The Co-Prop framework consists of two core modules: (1) Retrieval-Augmented Control Points Generation Module, which anchors key transition points in the audio, and (2) Audio-Insert Propagator, which propagates the segmentation frame-by-frame, integrating audio information to improve synchronization and reduce memory load. Evaluations on multiple datasets demonstrate better performance in alignment rates and segmentation precision than baseline models, particularly on multi-source audio benchmarks.

**Strengths:**

1. The use of a retrieval-augmented LLM approach to identify audio control points and address temporal misalignment is novel within the AVVS domain.
2. The proposed Co-Prop framework addresses a pain point in AVVS, which is useful for content creation in complex audio-visual environments like AR or video editing.
3. Experimental results show improved alignment rates, especially in scenarios with multiple sound sources, demonstrating the method’s efficacy in AVVS.
4. The proposed method is well presented, with comprehensive diagrams and examples illustrating the misalignment issues in prior models.

**Weaknesses:**

1. I am concerned that the paper does not explore how segmentation results would vary with different audio types. For instance, if a continuous dog bark is replaced with an intermittent one, it is unclear if the dog’s mask would disappear during pauses in barking. This scenario tests the model's adaptability to temporal gaps in sound, which is vital to confirm its robustness in handling real-world examples.
2. The paper would benefit from further experiments addressing complex audio scenarios, such as overlapping sounds or sounding objects that are off-screen. These situations are common in real-world settings, and I am curious if Co-Prop could maintain object integrity in such cases. If the model relies heavily on visual input alone when multiple audio cues overlap or when sounds lack visual sources, it may risk collapsing or misinterpreting segments, which could compromise segmentation quality.
3. The multi-step retrieval process in the RCPG module could be explained more clearly. For instance, while the ablation study in Table 2(b) shows performance improvements with 3-step prompts, it is not that clear why these prompts outperform simpler versions. I would encourage the authors to clarify how retrieval samples are chosen and if any cases show weaknesses in control point detection, as this would provide a clearer view of RCPG’s reliability.
4. While the authors report strong results on the evaluated datasets, I would like to see tests on additional in-the-wild audio-visual data to better gauge Co-Prop’s robustness. Applying Co-Prop to less curated datasets could validate its claims of temporal alignment across different audio contexts.
5. Although the paper mentions memory efficiency, no concrete results are provided to quantify these improvements. Memory usage is critical in AVVS applications, especially for long videos, and I would suggest including a direct comparison of memory consumption against baselines.
6. I am concerned that Qwen’s performance in detecting precise transition points may vary, as language models like Qwen are not specifically optimized for detecting fine-grained audio transitions. I recommend using acoustic event detectors, such as PANNs [1] or BEATs [2], which could potentially enhance the accuracy of detecting key audio transition points. These models are trained to recognize audio events and may provide more robust control point identification, leading to more consistent segmentation performance, especially in scenarios with overlapping sounds.

References

[1] Kong et al. PANNs: Large-Scale Pretrained Audio Neural Networks for Audio Pattern Recognition.

[2] Chen et al. BEATs: Audio Pre-Training with Acoustic Tokenizers.

**Questions:**

See weaknesses. I'd be happy to increase my rating if the authors address the weaknesses.

---

### Official Review · Reviewer_wfsZ · 2024-11-03

**Soundness:** 3
**Presentation:** 4
**Contribution:** 3
**Rating:** 5
**Confidence:** 4

**Summary:**

An AVS model to solve the temporal misalignment.

**Strengths:**

1. The paper intends to solve an important problem in AVS.

2. The visualization and representation are relatively clear.

**Weaknesses:**

1. Temporal misalignment: It seems an important problem to me, but the author should have expressed this specific problem more clearly. How many failure cases are caused by temporal misalignment? Are there any ratios? Is there any quantitative analysis beyond the qualitative cases? Additionally, the impact of temporal misalignment in Figure 5 is not that clear. In my perspective, most of the cases in Figure 5 are caused by simply incapable segmentation networks.

2. Missing important references and comparison: The major CV conferences, including CVPR, ICCV, and ECCV, have already published numerous papers on supervised AVS. However, the author does not compare any of these top works in Table 1. Here is a list of papers: [1-8].

3. Propagation process: Ablation on Audio-insert Propagator only tests with 4 layers and 1 layer. The natural process involves testing more settings, selecting the best, and reporting the peak performance. Would it achieve better results if using 8 layers?

4. Low amount of testing set: MOC (Multiple-sound Source Conversion) only contains 17 cases. Can these 17 cases serve as solid proof of the performance? I think it requires more data.

5. Accumulation error of key frame result: The model appears to rely on the segmentation result of the first frame. What if it is incorrect?

6. Temporal misalignment: In previous models like TPAVI and AVSegformer, there is no specific temporal information included in the model. It would be better to compare it with other models with temporal perception.

7. Some important AVS works in the related work: Works [10-12], including unsupervised/weak-supervised AVS and open-vocabulary AVS, need to be discussed in the related work section.

[1] (Cited but not compared) Chen, Y., Liu, Y., Wang, H., Liu, F., Wang, C., Frazer, H., & Carneiro, G. (2024). Unraveling Instance Associations: A Closer Look for Audio-Visual Segmentation. In Proceedings of the IEEE/CVF Conference on Computer Vision and Pattern Recognition (pp. 26497-26507).

[2] Chen, Y., Wang, C., Liu, Y., Wang, H., & Carneiro, G. (2024). CPM: Class-conditional Prompting Machine for Audio-visual Segmentation. arXiv preprint arXiv:2407.05358.

[3] Ma, J., Sun, P., Wang, Y., & Hu, D. (2024). Stepping stones: A progressive training strategy for audio-visual semantic segmentation. arXiv preprint arXiv:2407.11820.

[4] Hao, D., Mao, Y., He, B., Han, X., Dai, Y., & Zhong, Y. (2024, March). Improving audio-visual segmentation with bidirectional generation. In Proceedings of the AAAI Conference on Artificial Intelligence (Vol. 38, No. 3, pp. 2067-2075).

[5] Yan, S., Zhang, R., Guo, Z., Chen, W., Zhang, W., Li, H., ... & Gao, P. (2024, March). Referred by multi-modality: A unified temporal transformer for video object segmentation. In Proceedings of the AAAI Conference on Artificial Intelligence (Vol. 38, No. 6, pp. 6449-6457).

[6] (Cited but not compared) Yang, Q., Nie, X., Li, T., Gao, P., Guo, Y., Zhen, C., ... & Xiang, S. (2024). Cooperation Does Matter: Exploring Multi-Order Bilateral Relations for Audio-Visual Segmentation. In Proceedings of the IEEE/CVF Conference on Computer Vision and Pattern Recognition (pp. 27134-27143).

[7] Sun, P., Zhang, H., & Hu, D. (2024). Unveiling and Mitigating Bias in Audio Visual Segmentation. arXiv preprint arXiv:2407.16638.

[8] Nguyen, K. B., & Park, C. J. (2024). SAVE: Segment Audio-Visual Easy way using Segment Anything Model. arXiv preprint arXiv:2407.02004.

[9] Li, J., Yu, S., Wang, Y., Wang, L., & Lu, H. (2024, October). SelM: Selective Mechanism based Audio-Visual Segmentation. In Proceedings of the 32nd ACM International Conference on Multimedia (pp. 3926-3935).

[10] Liu, J., Liu, Y., Zhang, F., Ju, C., Zhang, Y., & Wang, Y. (2024). Audio-Visual Segmentation via Unlabeled Frame Exploitation. In Proceedings of the IEEE/CVF Conference on Computer Vision and Pattern Recognition (pp. 26328-26339).

[11] Liu, J., Wang, Y., Ju, C., Ma, C., Zhang, Y., & Xie, W. (2024). Annotation-free audio-visual segmentation. In Proceedings of the IEEE/CVF Winter Conference on Applications of Computer Vision (pp. 5604-5614).

[12] Guo, R., Qu, L., Niu, D., Qi, Y., Yue, W., Shi, J., ... & Ying, X. (2024). Open-Vocabulary Audio-Visual Semantic Segmentation. arXiv preprint arXiv:2407.21721.

I will consider raising my score if the authors can address the questions above.

**Questions:**

See the weaknesses above.

---

### Official Review · Reviewer_ERyZ · 2024-11-03

**Soundness:** 1
**Presentation:** 1
**Contribution:** 3
**Rating:** 3
**Confidence:** 4

**Summary:**

This paper focusses on the task of audiovisual video segmentation: given a sounding video, generate pixel-level maps of sound-producing objects that align with the ongoing audio.

Existing methods suffer from poor temporal alignement.

To tackle this issues the authors introduce a two-steps framework:
- LLM-assisted audio event segmentation.
- Segment-based downstream video segmentation.

The paper also proposes a new dataset and benchmark.

**Strengths:**

- The core novelty of this paper is the proposal to rely on audio segmentation to decompose the audio-visual video segmentation task into sub-problems.

**Weaknesses:**

- Although the high-level task description is clear, the paper lacks a deeper explanation of the workflow with simple terms (it is unclear whether the task is supervised, unsupervised or semi-supervised). The introduction dives into convoluted acronyms (e.g. Retrieval-augmented Control Points Generation Module (RCPG)) took quickly without telling what they are actually meant to achieve (detecting keyframes + reference masks). This makes the paper hard to understand.
  - I would suggest adding an overview of the task setup in the introduction (there is no description of the Figure 1 anywhere in the paper).
- RCPG is meant to perform audio segmentation, it should be treated as such and compared with other audio segmentation methods.
- Results are presented before metrics and data.
- The training protocol for the Keyframe Processor is unclear.

**Questions:**

- Is the task or audio-visual video segmentation supervised or unsupervised? (e.g. do have ground truth video segmentation pixel maps for training or not?)
  - There is mention of a semi-supervised approach in the related works, is this paper following the same approach? Please consider adding such detail.
- In eq. 1, what prompt is x? Why is a prompt needed at all? How do you mathematically define an audio frame? Is it a spectrogram?
- Is the training of Keyframe Mask Generation supervised, unsupervised or semi-supervised? I am reading that it is fine-tuned on keyframes extracted from the training dataset by RCPG, fine-tuned to do what exactly?
  - Please consider adding a step-by-step description of the training process.
- What do the scores in the Table 1 correspond to?
- Section 4.3: what is the "cosine similarity" used for? Please consider more details about its use.
- What happens when sounding event are overlapping? Please discuss how your method handles overlapping sound events, or if this is a limitation of your approach.

---

### Note · Authors · 2024-11-23

I have read and agree with the venue's withdrawal policy on behalf of myself and my co-authors.